# A high rate of COVID-19 vaccine hesitancy in a large-scale survey on Arabs

Eyad A Qunaibi[1]*, Mohamed Helmy[2,3], Iman Basheti[4], Iyad Sultan[5,6]

[1]Faculty of Pharmacy, Jerash University, Jerash, Jordan; [2]Computational Biology and Omics, Bioinformatics Institute (BII), Agency for Science, Technology and Research (A*STAR), Biopolis, Singapore, Singapore; [3]Department of Computer Science, Lakehead University, Thunder Bay, Canada; [4]Department of Clinical Pharmacy and Therapeutics, Faculty of Pharmacy, Applied Science Private University, Amman, Jordan; [5]Department of Paediatrics, University of Jordan, Amman, Jordan; [6]Department of Paediatrics and Cancer Care Informatics Program, King Hussein Cancer Center, Amman, Jordan

## Abstract

**Background:** Vaccine hesitancy can limit the benefits of available vaccines in halting the spread of COVID-19 pandemic. Previously published studies paid little attention to Arab countries, which has a population of over 440 million. In this study, we present the results of the first large-scale multinational study that measures vaccine hesitancy among Arab-speaking subjects.

**Methods:** An online survey in Arabic was conducted from 14 January 2021 to 29 January 2021. It consisted of 17 questions capturing demographic data, acceptance of COVID-19 vaccine, attitudes toward the need for COVID-19 vaccination and associated health policies, and reasons for vaccination hesitancy. R software v.4.0.2 was used for data analysis and visualization.

**Results:** The survey recruited 36,220 eligible participants (61.1% males, 38.9% females, mean age 32.6 ± 10.8 years) from all the 23 Arab countries and territories (83.4%) and 122 other countries (16.6%). Our analysis shows a significant rate of vaccine hesitancy among Arabs in and outside the Arab region (83% and 81%, respectively). The most cited reasons for hesitancy are concerns about side effects and distrust in health care policies, vaccine expedited production, published studies and vaccine producing companies. We also found that female participants, those who are 30–59 years old, those with no chronic diseases, those with lower level of academic education, and those who do not know the type of vaccine authorized in their countries are more hesitant to receive COVID-19 vaccination. On the other hand, participants who regularly receive the influenza vaccine, health care workers, and those from countries with higher rates of COVID-19 infections showed more vaccination willingness. Interactive representation of our results is posted on our project website at https://mainapp.shinyapps.io/CVHAA.

**Conclusions:** Our results show higher vaccine hesitancy and refusal among Arab subjects, related mainly to distrust and concerns about side effects. Health authorities and Arab scientific community have to transparently address these concerns to improve vaccine acceptance.

**Funding:** This study received no funding.

**\*For correspondence:**
eyad.aqunaibi@jpu.edu.jo

**Competing interests:** The authors declare that no competing interests exist.

## Introduction

It has been recognized early that the race to produce COVID-19 vaccines will not halt the pandemic unless there is a general acceptance by the public to take the vaccine (**Neumann-Böhme et al., 2020**; **Burgess et al., 2021**). Therefore, COVID-19 vaccination hesitancy has been studied heavily before and since the early stage of vaccine availability, with high variation in the willingness to be vaccinated among different communities (**Sallam, 2021**; **Feleszko et al., 2021**; **Lin et al., 2020**).

Several large multinational studies on vaccine acceptance have been conducted in 19 countries (n = 13,426) (*Lazarus et al., 2021*), 15 countries (n = 18,526) (*Ipsos, 2020*), 15 countries (n = 13,500) (*Mega, 2021*), 14 countries (n = 12,777) (*Ipsos, 2021*), and seven countries (n = 7662) (*Neumann-Böhme et al., 2020*). Interestingly, none of which covered Arabic-speaking nations. A smaller multinational study that surveyed 3414 participants was conducted in Jordan and Kuwait, with minor participation from some other Arab countries (*Sallam et al., 2021*). This study showed that vaccine acceptance was low (29.4%) and was lower in females, individuals with lower academic education, and individuals with no chronic diseases.

Other studies have been conducted in Saudi Arabia (n = 1000) and (n = 3101) (*Al-Mohaithef and Padhi, 2020*; *Magadmi and Kamel, 2020*), Egypt (n = 559) (*Abdelhafiz et al., 2020*), Jordan (n = 3100) (*El-Elimat et al., 2020*), and the UAE (n = 1109) (*Muqattash et al., 2020*). With the Arab nations having significant variations socioeconomically, politically and in the measures taken to control the pandemic, the study of reactions to and acceptance of the vaccine becomes necessary. Also, the authorization of the use of Sinopharm vaccine by some Arab countries, despite the lack of sufficient safety and effectiveness evidence (*Cyranoski, 2020*), may have an impact on the public's trust in the vaccine and the health policies in these countries. Furthermore, attitudes toward the vaccines are affected by complex and dynamic interplaying factors, and considerable changes over time have been observed in acceptance and hesitancy rates (*Lazarus et al., 2021*; *Ipsos, 2020*; *Mega, 2021*; *Ipsos, 2021*; *Wang et al., 2021*; *Imperial College London, 2021*; *Lewis, 2020*). For all of these reasons, the earlier local studies cannot be generalized to the Arab world, and further larger studies will present a clearer picture of the region.

Arab countries and territories (23 in total) span a large geographical area in North Africa and West Asia with a population of over 440 million (*Worldometer, 2021*). The total reported number of COVID-19 cases in the region until the mid-February 2021 was more than 4.1 million with 70.7 thousand deaths (*Dong et al., 2020*; *Appendix 1—figure 1*). Yet, the Arab region is understudied, despite the geographical spread, the number of residents, and the number of cases and deaths. So, a large-scale multinational study for this area is necessary.

Our study aims to fill the gaps by investigating vaccine acceptance using a large-scale survey targeting the relatively understudied Arab populations living in different countries around the world following vaccine availability and administration. Secondly, to unveil the barriers leading to vaccine hesitancy and their prevalence among the participants using an extensive updated list of barriers against vaccine acceptance. Thirdly, the study compares the answers of the respondents residing in and outside the Arab world to evaluate the effect of socioeconomic, cultural, health policies and political differences on their reported attitudes and barriers to acceptance.

## Materials and methods

The Survey of Arab COVID-19 Vaccine Acceptance (SACVA) is an open online survey that was conducted using the online platform https://www.surveyplanet.com/ from 14 January 2021 to 29 January 2021. The sample population was a convenience sample targeted through a digital campaign using social media platforms. Institutional Review Board (IRB) approval was obtained from the last author's institution. Unique IP addresses are allowed to participate once on the survey platform to prevent multiple entries. Consent to participate was obtained at the first entry of the survey portal for each participant. The platform allows participants to move through screens only when answers were obtained, which prevents missing entries. The survey consisted of 17 questions, including the consent to participate. All questions were written and validated in the Arabic language – an English translation of the questions can be found on *Supplementary file 1*. Questions two to nine captured demographics and current health status; question 10 was about the annual influenza vaccine; question 11 was about available vaccine(s) in each country (if known), and answers to subsequent questions were directed based on the type of available vaccine(s). Questions 12 and 13 queried whether the participant received the COVID-19 vaccination and if they had any side effects. Those who had already taken the vaccine were not allowed to answer question 14 that queried participant's acceptance/hesitance toward COVID-19 vaccine; these participants were not included in the analysis reported in this paper. Questions 15 and16 surveyed participants' attitudes toward the need for COVID-19 vaccination and associated health policies. Question 17 was a detailed question that evaluated 29 barriers, which potentially influenced the decision to receive the vaccine in addition to 'I do

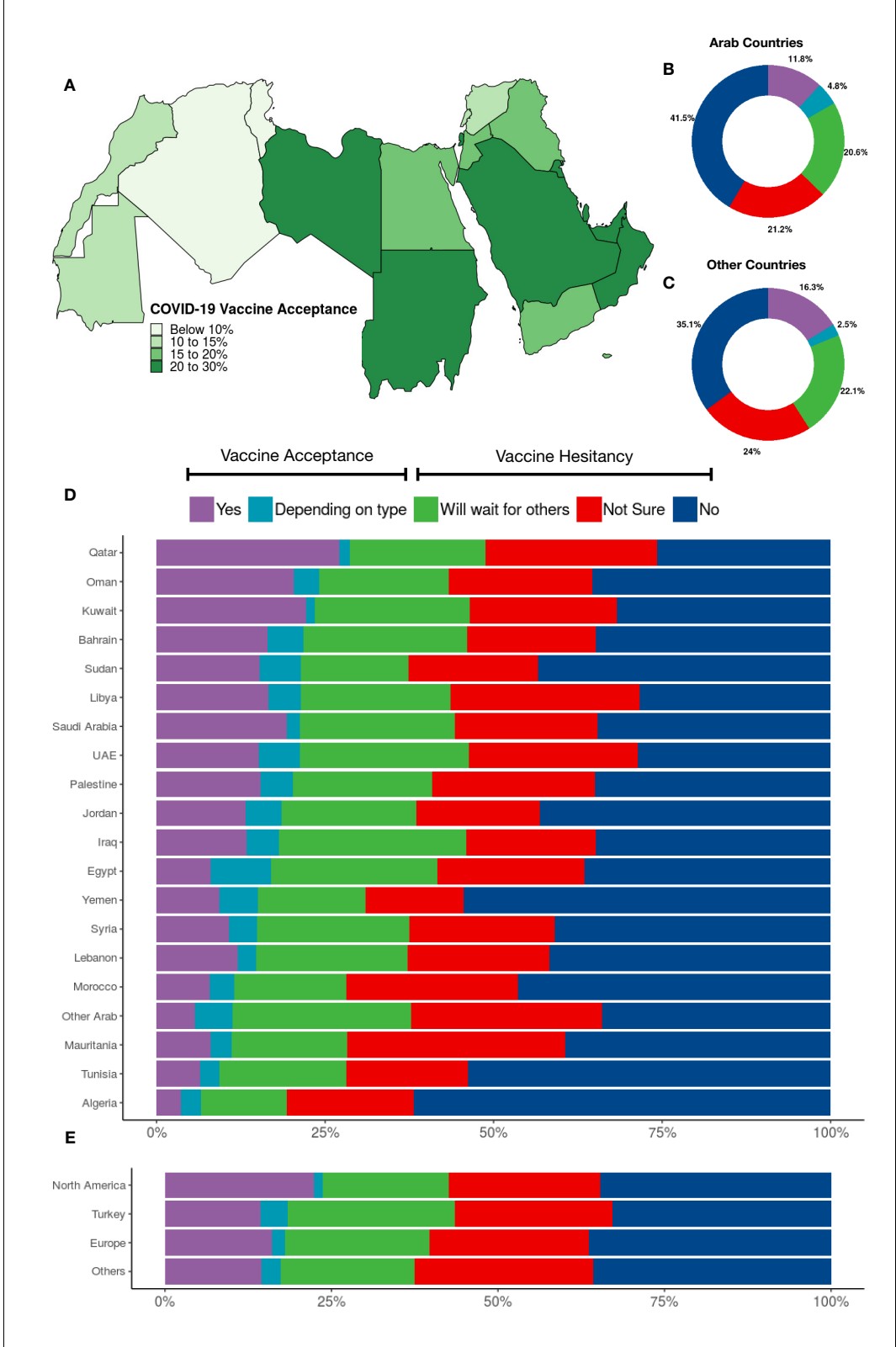

**Figure 1.** COVID-19 vaccination attitudes among 36,220 participants. (**A**) Vaccine acceptance in the per-country in the Arab region, (**B**) vaccination attitudes reported by participants from the Arab countries and territories, (**C**) vaccination attitudes reported by participants from countries other than Arab countries and territories, (**D**) vaccination attitudes reported by participants per Arab country/territory, and (**E**) vaccination attitudes reported by participants from countries other than Arab countries and territories clustered by residency region.

not have any reservations about taking the vaccine' option. We also allowed participants who answered 'Yes', meaning they are willing to be vaccinated, to choose from the 29 barriers.

Questions were discussed thoroughly among authors and other colleagues. Face validity was tested by the third author, who has expertise in the domain. A pilot survey was then posted online and 100 individuals participated following direct contacts by authors. Analyzing responses and comments of this pilot survey helped in refining the final survey and confirming its validity and reliability.

The survey data was analyzed using R software v.4.0.2. Descriptive statistics and analytical graphs were used as needed. Participants were also subcategorized based on country of residence. Arab countries with less than 100 participants (Somalia, Djibouti, and Comoros) were grouped together in one category and labeled 'Other Arab countries'. The non-Arab countries where the Arabic-speaking respondents were residing were classified into groups: European countries (n = 30), North American countries (n = 3), Turkey, and the rest of non-Arab countries as others (n = 88). The answers to the 14th question, 'Do you intend to take the vaccine?', were used as a dependent variable and were analyzed using binary logistic regression. Two of the answer choices ('Yes', 'Depends on the type of vaccine') were used to define vaccine acceptance, while the other three ('No', 'Not sure', 'I will wait and see its effects on others') were labeled as 'Vaccine Hesitancy'. Responses to the question of the barriers to acceptance (Question 17) were compared for gender, academic education, and country of residence using chi-square. Our acquisition and analysis of the results followed the guidelines of the CHERRIES checklist (*Eysenbach, 2004*).

As for COVID-19 cases and death statistics, we used the COVID19 package v2.3.2 that collects data from different sources to provide up-to-date COVID-19 statistics (*Guidotti and Ardia, 2020*). The total number of confirmed cases and deaths were correlated with vaccine acceptance in different Arab countries using the Spearman correlation. The results of the survey can be found on the project's website at https://mainapp.shinyapps.io/CVHAA, while the data and the R code written for the analysis can be found on the project's GitHub repository (*Qunaibi, 2021*).

## Results

### Demographics

Our online survey raw data were downloaded on 29 January 2021; there were 38,485 participants who started filling the survey of whom 36,958 consented and proceeded with the survey. A total of 738 participants reported receiving COVID-19 vaccination before filling the survey and were excluded from further analysis in this report bringing the total respondents who qualify for analysis to 36,220. The participants cover all the 23 Arab countries and territories (n = 30,200, 83.4%) and Arabs who live in 122 other countries (n = 6020, 16.6%). Participants from countries out of the Arab region were clustered into four groups: Europe (N = 3130, 52%), North America (n = 748, 12.4%), Turkey (n = 1630, 27.1%), and others (n = 512, 8.5%).

The mean age was 32.6 years (±10.8). There were more males (n = 22,040, 61.1%) than females (n = 14,180, 38.9%) – *Appendix 1—figure 2*. Chronic diseases were reported by 5839 participants (16.1%). Previous COVID-19 infection – suspected or confirmed – was reported by 6637 (18.3%) participants; 11,458 (31.6%) other participants were not sure if they had contracted the virus. Among the 4494 participants who reported testing for COVID-19, there were 2792 participants with positive test results (62.1% positivity). Only 908 (2.5%) participants reported annual influenza vaccine, while 28,040 (77.4%) reported never receiving it. More than half of the participants had a bachelor's degree or higher (22,236, 61.4%). Being a health care worker (HCW) was reported by 5708 participants (15.8%). When asked about the type of vaccine available in their countries, 15,057 (41.6%) did not know the type, while vaccines made in China and the United States were reported by 12,374 and 12,254 participants, respectively. Detailed participant characteristics are shown in *Table 1*.

### COVID-19 vaccination hesitancy and related factors

When asked about their willingness to receive COVID-19 vaccine if the option is available to them, 4548 (12.6%) of the respondents answered 'Yes'; 1615 (4.5%) answered 'Depends on the type of vaccine'; 7552 (20.9%) answered 'I will wait and see its effects on others'; 7856 (21.7%) answered 'I am not sure'; and 14,649 (40.4%) chose 'No'. The first two choices were considered acceptance to

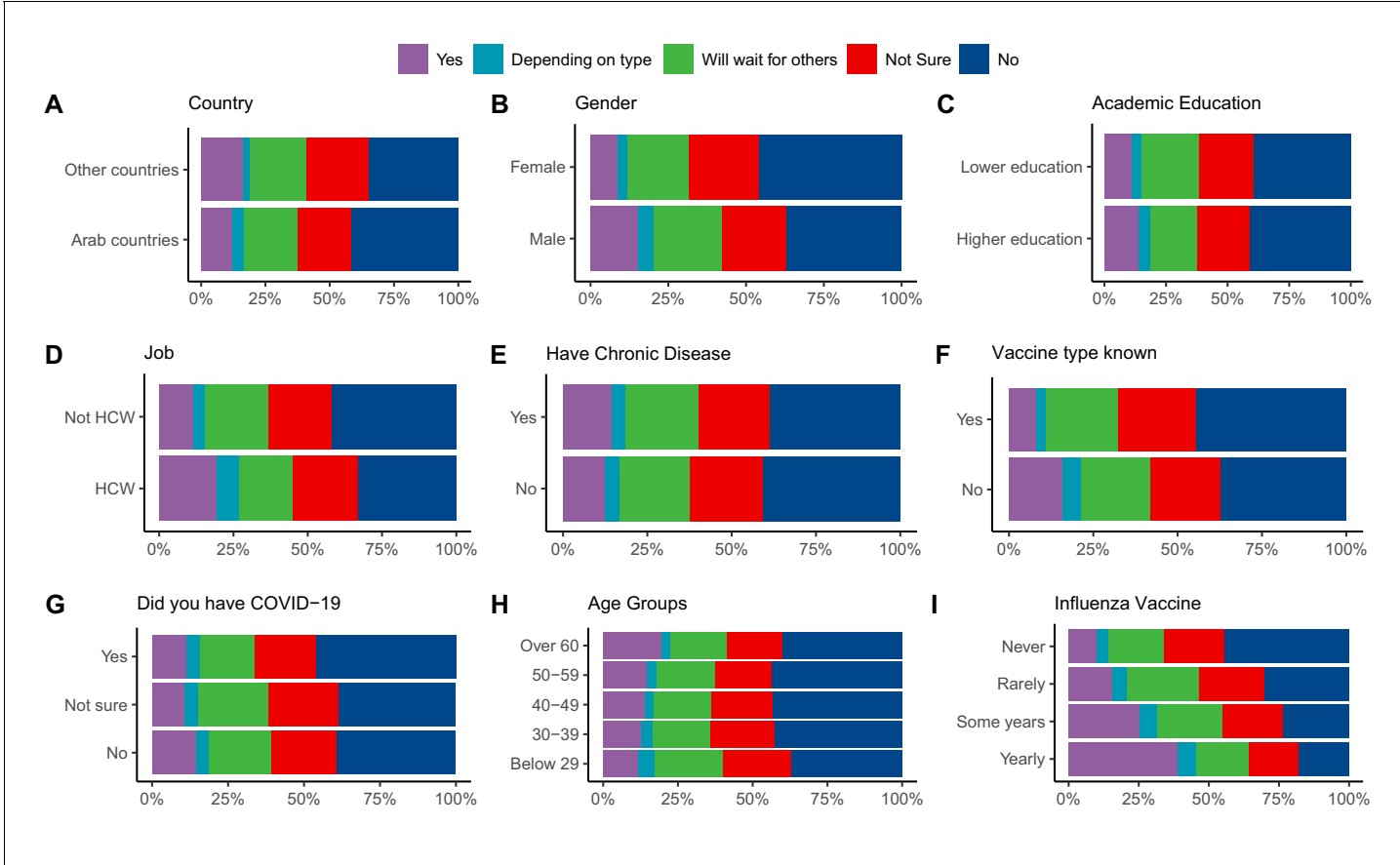

**Figure 2.** Differences in COVID-19 vaccination attitudes among participants according to (A) country of residence, (B) age, (C) level of academic education, (D) being a health care worker, (E) having a chronic illness, (F) knowing the vaccine type available in participant's country, (G) having a previous COVID-19 infection, (H) age, and (I) receiving annual influenza vaccine.

receive a vaccine, while the last three were labeled as vaccine hesitancy (*Figure 1*). Variations in responses were analyzed using different factors as covariates (*Figure 2*).

Respondents from the Arab Gulf countries (Qatar, Oman, Kuwait, Bahrain, Saudi Arabia, and UAE) plus Libya and Sudan showed the highest willingness for vaccination, while those who showed the least willingness are participants from the west region (Algeria, Tunisia, Mauritania, and Morocco) (*Figure 1*). Arabic-speaking participants living in North America were more willing to receive vaccination than those in the other three clusters (*Figure 1*, *Table 2*).

Several factors (shown in *Figure 3*, *Table 3*) were tested in a binomial logistic regression model to examine their correlation with vaccine hesitancy. Univariate and multivariate analyses showed that almost all tested factors were significant predictors for vaccine hesitancy, reflecting the large sample size tested. Odds ratio (OR) showed the stronger effect of the following factors on participants' hesitance: Never (OR, 4.04) or rarely (OR, 2.69) receiving the influenza vaccine, not knowing the vaccine type available (OR, 1.93), female gender (OR, 1.91), and outside of the health care system (OR, 1.84). Vaccine acceptance in each Arab country was correlated with the number of confirmed COVID-19 cases and deaths using Spearman correlation. It was found out that the number of cases (p=0.0047) but not deaths (p=0.3) correlated significantly with vaccine acceptance (*Appendix 1—figure 3*).

## Barriers to acceptance

There were 3905 participants who chose acceptance but yet had one or more barrier(s) selected. Of the 29 barriers, the most common responses were 'I am afraid side effects of the vaccine will develop, other than what has been disclosed' – 22,235 (61.4%), 'Not enough time has passed to

**Table 1.** Characteristics of participants with distribution of COVID-19 vaccine willingness.

| Variables | Levels | No | | Not sure | | Will wait for others | | Depending on type | | Yes | |
|---|---|---|---|---|---|---|---|---|---|---|---|
| | | N | % | N | % | N | % | N | % | N | % |
| Age | Below 29 | 5871 | 37.2 | 3607 | 22.9 | 3591 | 22.8 | 899 | 5.7 | 1803 | 11.4 |
| | 30–39 | 4939 | 42.7 | 2484 | 21.5 | 2225 | 19.2 | 463 | 4.0 | 1454 | 12.6 |
| | 40–49 | 2563 | 43.3 | 1213 | 20.5 | 1156 | 19.5 | 160 | 2.7 | 827 | 14.0 |
| | 50–59 | 1009 | 43.9 | 429 | 18.6 | 455 | 19.8 | 72 | 3.1 | 336 | 14.6 |
| | Over 60 | 267 | 40.2 | 123 | 18.5 | 125 | 18.8 | 21 | 3.2 | 128 | 19.3 |
| Chronic Diseases | No | 12,390 | 40.8 | 6614 | 21.8 | 6292 | 20.7 | 1368 | 4.5 | 3717 | 12.2 |
| | Yes | 2259 | 38.7 | 1242 | 21.3 | 1260 | 21.6 | 247 | 4.2 | 831 | 14.2 |
| Country | Arab countries | 12,534 | 41.5 | 6414 | 21.2 | 6220 | 20.6 | 1464 | 4.8 | 3568 | 11.8 |
| | Other countries | 2115 | 35.1 | 1442 | 24.0 | 1332 | 22.1 | 151 | 2.5 | 980 | 16.3 |
| Academic Education | Higher education | 9128 | 41.1 | 4752 | 21.4 | 4277 | 19.2 | 1027 | 4.6 | 3052 | 13.7 |
| | Lower education | 5521 | 39.5 | 3104 | 22.2 | 3275 | 23.4 | 588 | 4.2 | 1496 | 10.7 |
| Had Covid | No | 7147 | 39.4 | 3885 | 21.4 | 3703 | 20.4 | 780 | 4.3 | 2610 | 14.4 |
| | Not sure | 4445 | 38.8 | 2628 | 22.9 | 2642 | 23.1 | 548 | 4.8 | 1195 | 10.4 |
| | Yes | 3057 | 46.1 | 1343 | 20.2 | 1207 | 18.2 | 287 | 4.3 | 743 | 11.2 |
| Job | HCW | 1886 | 33.0 | 1266 | 22.2 | 1034 | 18.1 | 432 | 7.6 | 1090 | 19.1 |
| | Not HCW | 12,763 | 41.8 | 6590 | 21.6 | 6518 | 21.4 | 1183 | 3.9 | 3458 | 11.3 |
| Gender | Male | 8152 | 37.0 | 4625 | 21.0 | 4776 | 21.7 | 1152 | 5.2 | 3335 | 15.1 |
| | Female | 6497 | 45.8 | 3231 | 22.8 | 2776 | 19.6 | 463 | 3.3 | 1213 | 8.6 |
| Influenza vaccine | Yearly | 166 | 18.3 | 159 | 17.5 | 170 | 18.7 | 62 | 6.8 | 351 | 38.7 |
| | Some years | 687 | 23.6 | 627 | 21.5 | 678 | 23.3 | 180 | 6.2 | 739 | 25.4 |
| | Rarely | 1324 | 30.4 | 1019 | 23.4 | 1117 | 25.6 | 228 | 5.2 | 673 | 15.4 |
| | Never | 12,472 | 44.5 | 6051 | 21.6 | 5587 | 19.9 | 1145 | 4.1 | 2785 | 9.9 |
| Vaccine type unknown | No | 7937 | 37.5 | 4356 | 20.6 | 4364 | 20.6 | 1141 | 5.4 | 3365 | 15.9 |
| | Yes | 6712 | 44.6 | 3500 | 23.2 | 3188 | 21.2 | 474 | 3.1 | 1183 | 7.9 |

verify the vaccine's safety' – 20,172 (55.7%), 'The vaccine production has been rushed, making me doubt the credibility of the producing company' – 16,698 (46.1%), 'I do not trust the health care policies applied in my country' – 14,151 (39.1%), and 'I do not trust the published studies, nor the company producing the vaccine' – 11,968 (33%) (*Figure 4*).

## Comparison of participants inside and outside the Arab World

Participants in the Arab World were slightly more likely to have vaccine hesitancy when compared to those living outside (83.3% vs. 81.2%) (*Figure 1*). Those living in North America were the least hesitant (76.3%), while those living in Turkey had the highest hesitancy (83.6%). Additionally, participants living in Arab countries were more likely to report 'I do not trust the health care policies applied in my country', 'There are no published studies on the vaccine', 'I do not trust the published studies, nor the company producing the vaccine', 'No need for the vaccine as rates of viral infection are decreasing', and 'No need for the vaccine as most people in my country have already been infected' (chi-square test, p<0.0001, with difference >5% for all) (*supplementary file 2*, *Figure 4—figure supplement 1*).

## Attitudes toward vaccination policies and need

When asked about their opinions regarding suggested vaccination policies, participants' responses were to let people choose if they want to take it or not (59.5%); to mandate it on populations in which the vaccine was proven to be effective and safe as per clinical studies (13.6%), not sure

**Table 2.** List of surveyed countries and the frequency (%) of participants COVID-19 vaccination choices.

| Country | No | | Not sure | | Will wait for others | | Depending on type | | Yes | | Total |
|---|---|---|---|---|---|---|---|---|---|---|---|
| | N | % | N | % | N | % | N | % | N | % | |
| Algeria | 1675 | 61.9 | 509 | 18.8 | 343 | 12.7 | 81 | 3.0 | 98 | 3.6 | 2706 (7.5) |
| Bahrain | 40 | 34.8 | 22 | 19.1 | 28 | 24.3 | 6 | 5.2 | 19 | 16.5 | 115 (0.3) |
| Egypt | 1949 | 36.5 | 1166 | 21.8 | 1315 | 24.6 | 480 | 9.0 | 429 | 8.0 | 5339 (14.7) |
| Europe | 1139 | 36.4 | 749 | 23.9 | 677 | 21.6 | 61 | 1.9 | 504 | 16.1 | 586 (1.6) |
| Iraq | 204 | 34.8 | 113 | 19.3 | 163 | 27.8 | 28 | 4.8 | 78 | 13.3 | 7020 (19.4) |
| Jordan | 3032 | 43.2 | 1283 | 18.3 | 1407 | 20.0 | 369 | 5.3 | 929 | 13.2 | 529 (1.5) |
| Kuwait | 168 | 31.8 | 115 | 21.7 | 122 | 23.1 | 7 | 1.3 | 117 | 22.1 | 492 (1.4) |
| Lebanon | 205 | 41.7 | 104 | 21.1 | 110 | 22.4 | 14 | 2.8 | 59 | 12.0 | 229 (0.6) |
| Libya | 65 | 28.4 | 64 | 27.9 | 51 | 22.3 | 11 | 4.8 | 38 | 16.6 | 99 (0.3) |
| Mauritania | 39 | 39.4 | 32 | 32.3 | 17 | 17.2 | 3 | 3.0 | 8 | 8.1 | 3775 (10.4) |
| Morocco | 1750 | 46.4 | 961 | 25.5 | 631 | 16.7 | 135 | 3.6 | 298 | 7.9 | 187 (0.5) |
| North America | 259 | 34.6 | 170 | 22.7 | 142 | 19.0 | 10 | 1.3 | 167 | 22.3 | 53 (0.1) |
| Oman | 66 | 35.3 | 40 | 21.4 | 36 | 19.3 | 7 | 3.7 | 38 | 20.3 | 1624 (4.5) |
| Other Arabs | 18 | 34.0 | 15 | 28.3 | 14 | 26.4 | 3 | 5.7 | 3 | 5.7 | 443 (1.2) |
| Others | 183 | 35.7 | 137 | 26.8 | 103 | 20.1 | 15 | 2.9 | 74 | 14.5 | 3588 (9.9) |
| Palestine | 568 | 35.0 | 392 | 24.1 | 336 | 20.7 | 77 | 4.7 | 251 | 15.5 | 313 (0.9) |
| Qatar | 114 | 25.7 | 113 | 25.5 | 89 | 20.1 | 7 | 1.6 | 120 | 27.1 | 1232 (3.4) |
| Saudi Arabia | 1240 | 34.6 | 761 | 21.2 | 822 | 22.9 | 74 | 2.1 | 691 | 19.3 | 665 (1.8) |
| Sudan | 136 | 43.5 | 60 | 19.2 | 50 | 16.0 | 19 | 6.1 | 48 | 15.3 | 979 (2.7) |
| Syria | 504 | 40.9 | 266 | 21.6 | 279 | 22.6 | 51 | 4.1 | 132 | 10.7 | 226 (0.6) |
| Tunisia | 358 | 53.8 | 120 | 18.0 | 125 | 18.8 | 19 | 2.9 | 43 | 6.5 | 3130 (8.6) |
| Turkey | 534 | 32.8 | 386 | 23.7 | 410 | 25.2 | 65 | 4.0 | 235 | 14.4 | 748 (2.1) |
| UAE | 280 | 28.6 | 245 | 25.0 | 246 | 25.1 | 60 | 6.1 | 148 | 15.1 | 1630 (4.5) |
| Yemen | 123 | 54.4 | 33 | 14.6 | 36 | 15.9 | 13 | 5.8 | 21 | 9.3 | 512 (1.4) |

(10.9%), should not be given to anybody (6.1%); and to give work and transportation privileges to whomever takes the vaccine (3.9%). When asked who needs the vaccine, responses were as follows: whomever – the vaccine was proven to be effective and safe as per clinical studies (35.4%), specific categories of people need it, but they are not the majority (30.5%), I don't know (24.9%), and no one needs it (9.2%) (*Figure 5*).

## Discussion

This study presents the largest online survey on vaccine hesitancy that covered a heterogeneous population of Arabic people living all over the globe. In addition, it bridged the gap in knowledge on vaccine hesitancy in the Arab region. It shows low rates of vaccine acceptance in the face of the ongoing pandemic. Only one in eight respondents (12.5%) reported their willingness to take the vaccine. One in 22 (4.4%) based their decision on whether to take the vaccine or not on the type of the vaccine, acknowledging that the vaccine type they prefer may not be available in the near future and even when available, they may not be given the choice of selecting that vaccine type. These results are of unique significance because the study has been conducted after the vaccine has become available and administered to millions of people worldwide, and while about 70 different vaccine candidates are currently under development.

The study also showed a clear correlation between acceptance and gender, academic background, attitudes toward the flu shot, having been previously suspected of – or confirmed with – COVID-19 infection and knowledge of the vaccine type. Females were more hesitant to take the

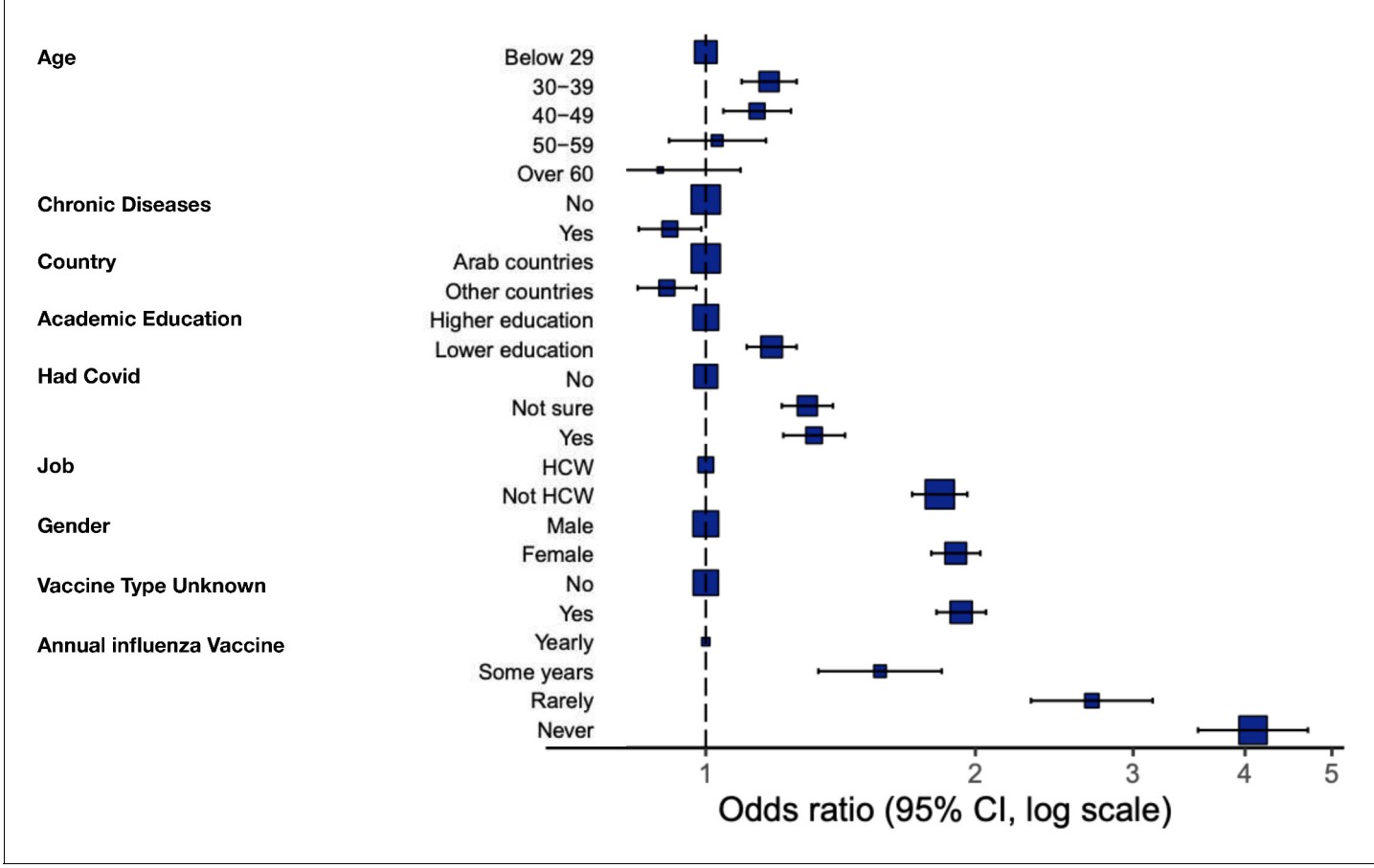

**Figure 3.** Multivariate analysis results of COVID-19 vaccine acceptance/hesitancy stratified according to different factors; odds ratio (OR) and 95% confidence intervals (CI) are shown; the size of the box represents the number of participants in each level.

vaccine, while the previous studies were inconclusive on the correlation of hesitance with gender, where women were found to have higher (*Fisher et al., 2020*; *Grech et al., 2020*; *Kreps et al., 2020*), equal (*Abdelhafiz et al., 2020*), or lower (*Lazarus et al., 2021*; *Ward et al., 2020*) hesitancy compared with men. Our results show lower acceptance in participants with current or previous suspected or confirmed COVID-19 infection (data) when compared with Lazarus et al. (n = 13,426) who found no significant correlation (*Lazarus et al., 2021*). On the other hand, our results are consistent with the literature in terms of lower acceptance in people who do not get influenza vaccination (*Lin et al., 2020*) and who have lower academic education (*Lin et al., 2020*; *Lazarus et al., 2021*).

Respondents who did not know the vaccine type available to them showed increased hesitancy. This may be attributed to the fact that some Arab countries were first to approve the Sinopharm vaccine despite lack of affirmative data (*Cyranoski, 2020*). The impact of vaccine efficacy on attitudes toward vaccination has been echoed in the study of Harapan et al. (n = 1359) (*Harapan et al., 2020*), conducted before vaccine availability, where 93.3% of respondents chose to be vaccinated with a 95% effective vaccine, but this acceptance rate decreased to 67.0% in the case of a vaccine with 50% effectiveness. The results showed a level of mistrust in health care policies in Arab countries (44%) which can also be attributed to the selection of certain vaccines, as well as the inability to choose which vaccine to take. All of these factors may contribute to high hesitance when the vaccine type is unknown to the participant.

Consistent with previous studies (*Harapan et al., 2020*; *Detoc et al., 2020*), HCW were more accepting the vaccine, although with still low proportions of about one in four (18% yes and 7.1% depending on the type). One study on Congolese HCW (n = 613) conducted in March–April, 2020 reported a similar notably low rate of acceptance (only 27.7%) (*Kabamba Nzaji et al., 2020*). In addition, consistent with the results of the multinational study by Lazarus et al. (n = 13,426)

**Table 3.** Predictors of vaccine hesitancy tested by univariate and multivariate binary logistic regression.

| Variable | Levels | Acceptance | Hesitance | Univariate | Multivariate |
|---|---|---|---|---|---|
| | | N (%) | N (%) | OR (95% CI) | OR (95% CI) |
| Age | Below 29 | 2702 (17.1) | 13,069 (82.9) | – | – |
| | 30–39 | 1917 (16.6) | 9648 (83.4) | 1.04 (0.98–1.11) | 1.18 (1.10–1.26) |
| | 40–49 | 987 (16.7) | 4932 (83.3) | 1.03 (0.95–1.12) | 1.14 (1.05–1.24) |
| | 50–59 | 408 (17.7) | 1893 (82.3) | 0.96 (0.86–1.08) | 1.03 (0.91–1.17) |
| | Over 60 | 149 (22.4) | 515 (77.6) | 0.71 (0.59–0.86) | 0.89 (0.73–1.09) |
| Chronic diseases | No | 5085 (16.7) | 25,296 (83.3) | – | – |
| | Yes | 1078 (18.5) | 4761 (81.5) | 0.89 (0.83–0.96) | 0.91 (0.84–0.99) |
| Country | Arab countries | 5032 (16.7) | 25,168 (83.3) | – | – |
| | Other countries | 1131 (18.8) | 4889 (81.2) | 0.86 (0.80–0.93) | 0.90 (0.84–0.98) |
| Academic education | Higher education | 4079 (18.3) | 18,157 (81.7) | – | – |
| | Lower education | 2084 (14.9) | 11,900 (85.1) | 1.28 (1.21–1.36) | 1.18 (1.11–1.26) |
| Had COVID | No | 3390 (18.7) | 14,735 (81.3) | – | – |
| | Not sure | 1743 (15.2) | 9715 (84.8) | 1.28 (1.20–1.37) | 1.30 (1.22–1.39) |
| | Yes | 1030 (15.5) | 5607 (84.5) | 1.25 (1.16–1.35) | 1.32 (1.22–1.43) |
| Job | HCW | 1522 (26.7) | 4186 (73.3) | – | – |
| | Not HCW | 4641 (15.2) | 25,871 (84.8) | 2.03 (1.90–2.17) | 1.82 (1.70–1.96) |
| Gender | Male | 4487 (20.4) | 17,553 (79.6) | – | – |
| | Female | 1676 (11.8) | 12,504 (88.2) | 1.91 (1.80–2.03) | 1.90 (1.79–2.03) |
| Vaccine type unknown | No | 4506 (21.3) | 16,657 (78.7) | – | – |
| | Yes | 1657 (11.0) | 13,400 (89.0) | 2.19 (2.06–2.33) | 1.93 (1.81–2.06) |
| Annual influenza vaccine | Yearly | 413 (45.5) | 495 (54.5) | – | – |
| | Some years | 919 (31.6) | 1992 (68.4) | 1.81 (1.55–2.11) | 1.57 (1.34–1.83) |
| | Rarely | 901 (20.7) | 3460 (79.3) | 3.20 (2.76–3.72) | 2.70 (2.31–3.15) |
| | Never | 3930 (14.0) | 24,110 (86.0) | 5.12 (4.47–5.86) | 4.08 (3.54–4.70) |

(*Lazarus et al., 2021*), participants from countries with higher per million COVID-19 cases were more likely to welcome the vaccine. Participants between the ages of 30 and 59 were less willing to receive vaccination compared with older participants, an expected result given the fact that COVID-19 severity is associated with older age. However, younger participants (<29 years old) showed more willingness to be vaccinated.

Among the 29 different reasons for vaccine rejection/hesitancy, the top two reasons selected by the respondents reflected concerns about safety, while the next three most prevalent reasons were issues of distrust. This is consistent with the literature, which showed high levels of distrust and concern about safety (*Lin et al., 2020*).

The three forms of distrust (in health care policies, in vaccine expedited production and in published studies) were notably higher among respondents residing in the Arab countries than those living outside the Arab world. The same applies to the belief that the vaccine has not been tested on a large enough number of people, just tens or hundreds, which reflects less awareness of the vaccine development process in the Arab countries and highlights the need to educate the general public on the subject. Similarly, more residents of the Arab world believe that the vaccine is not necessary anymore because most people in the participant's country 'have already been infected' or because the infection rate is decreasing. The infection rate is in fact decreasing (*Appendix 1—figure 1A*), but the public may need to be made aware that future outbreaks are still a possibility.

With the high rate of distrust, any form of coercion to take the vaccine may have negative impacts. Lazarus et al.'s large-scale study indicated that promoting voluntary acceptance is a better route and that coercion should be avoided (*Lazarus et al., 2021*). Similarly, a systematic review

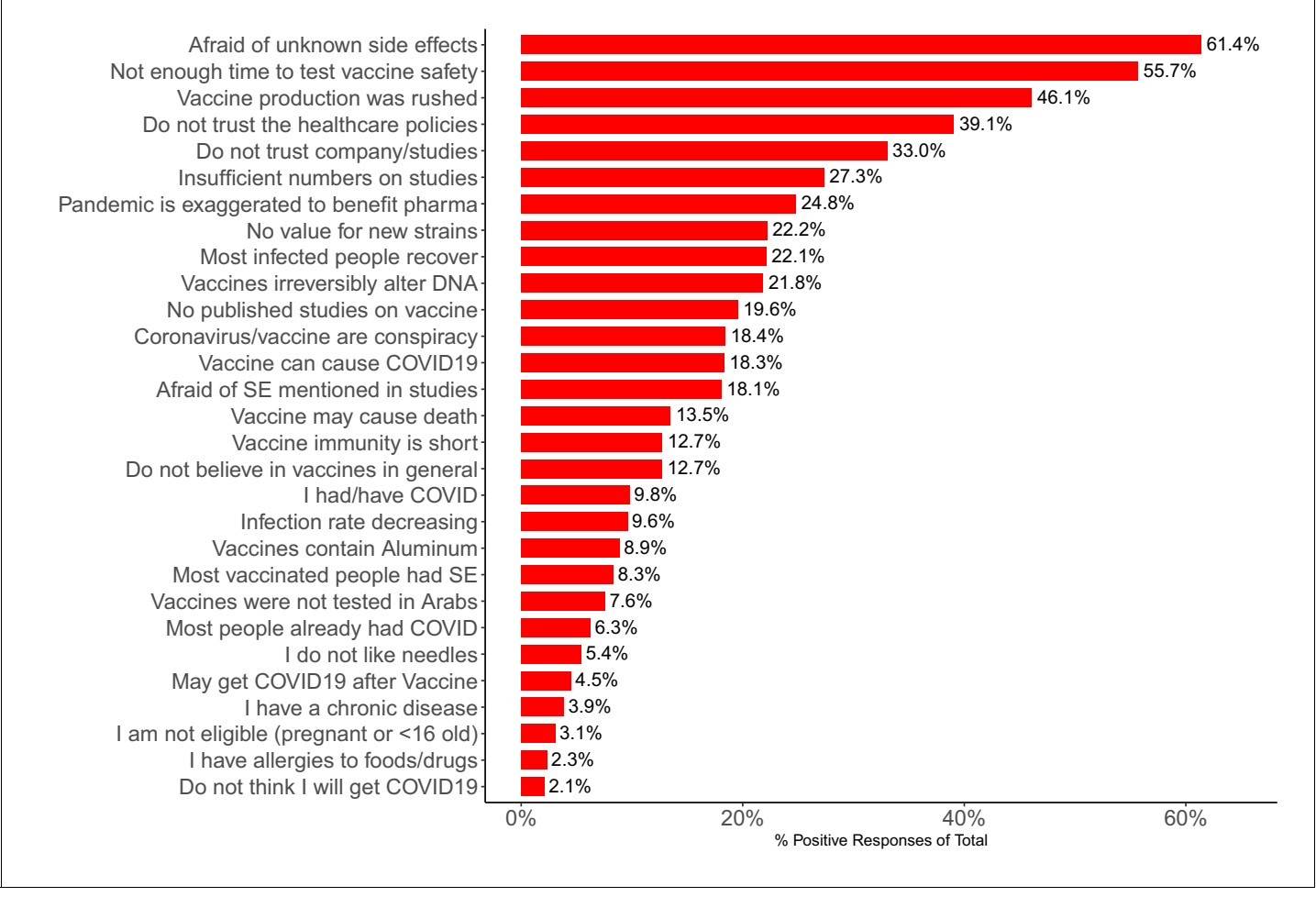

**Figure 4.** Barplot showing percentages of participants (N = 36,220) who selected the shown barriers.

The online version of this article includes the following figure supplement(s) for figure 4:

**Figure supplement 1.** Barriers to COVID-19 vaccine acceptance chosen by survey participants with percentage of selection for each barrier (out of a total of 36,220 participants) stratified according to (**A**) gender and (**B**) residence in or out of the Arab countries.

indicated that 'mandates could increase resistance' (*Lin et al., 2020*). In our study, the majority of participants (59.5%) believed that vaccination should be left to individual choices and only a minority believed it should be mandated for certain categories of people (6.1%) or on populations in which the vaccine was proven to be effective and safe as per clinical studies (13.6%).

Approximately one-fifth of our respondents chose 'The vaccine might lose its efficacy against the new viral strains' as a reason for hesitation. The survey was published shortly after reports of the new viral strains in the UK and South Africa have been made public, and with the recent reports of decreased efficacy of some vaccines (*Kabamba Nzaji et al., 2020*; *The Guardian, 2021*; *Knoll and Wonodi, 2021*), this concern of efficacy is expected to increase among the public.

Several factors appear to contribute to the low level of vaccine acceptance in the current study compared with the previous works. First of all, the response to the question of willingness is broken down from Yes/No or Likert scale (in previous works) to a spectrum of choices which could more accurately detect the hesitant respondents who could have otherwise chosen (Yes) or (Agree/ Strongly agree). In a large-scale survey conducted in October 2020 that included 18,526 adults across 15 countries (*Ipsos, 2020*), 73% strongly agreed or agreed that 'if a vaccine for COVID-19 were available, I would get it'. However, of those, only 22% agreed that they would become vaccinated 'immediately after the vaccine is available', while some others chose that they would wait for a year and even longer. The same study found out that there is less certainty about getting vaccinated

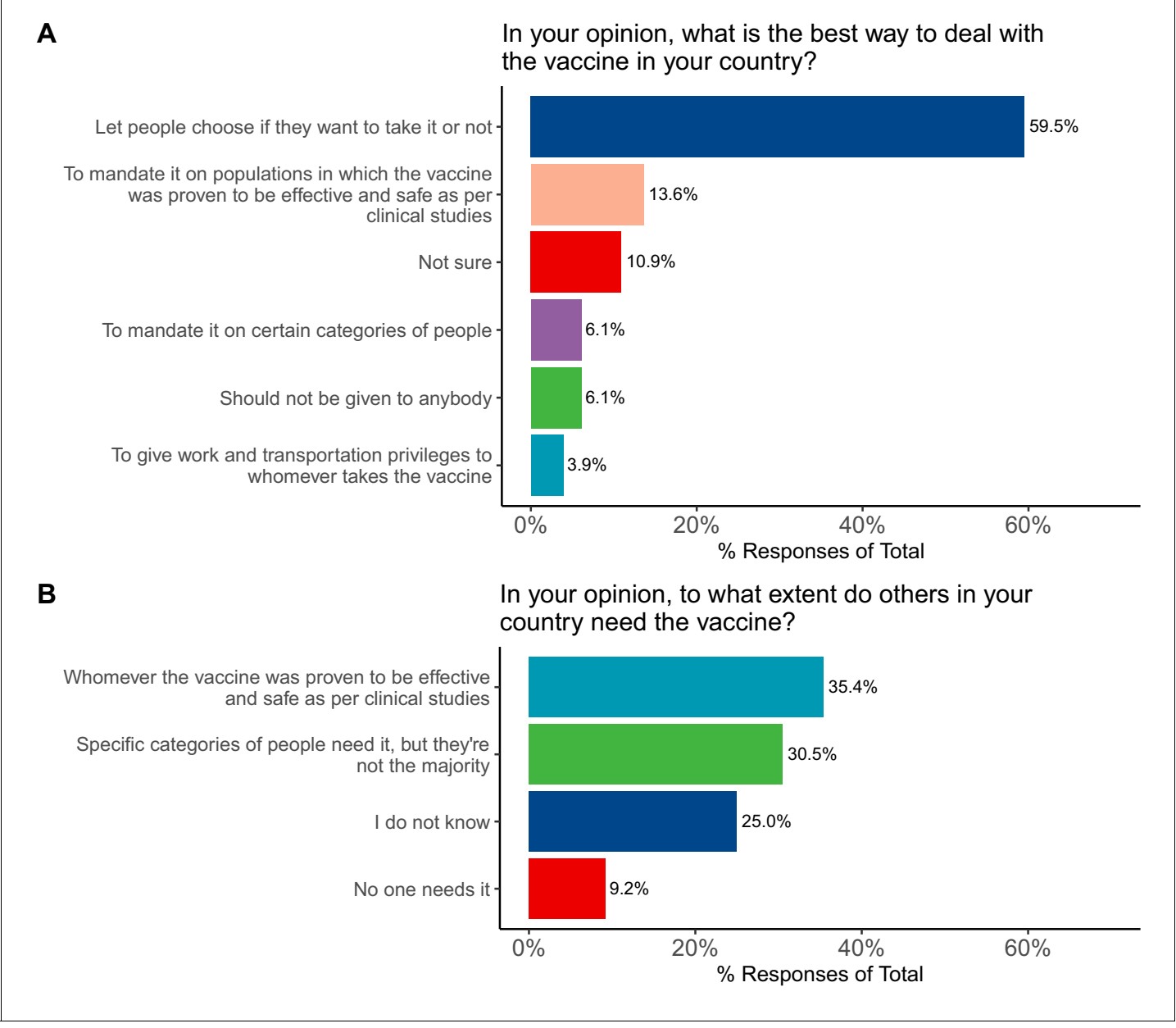

**Figure 5.** Participants' attitudes toward COVID-19 vaccination in regards to (**A**) national health policies and (**B**) selecting individuals who should be vaccinated.

among those who will wait. Thus, the affirmative nature of the Yes/No and 5-point Likert scales do not seem to reflect the true nature of hesitancy and whether or not it changes its nature over time.

In 2015, the World Health Organization (WHO) Strategic Advisory Group of Experts on Immunization defined vaccine hesitancy as a 'delay in acceptance or refusal of vaccination despite the availability of vaccination services' (*MacDonald, 2015*). We, therefore, question considering those who intend to take the vaccine after a prolonged time of availability as 'Accepting' since this may interfere with the targeted achievement of collective immunity.

In the present study, 20.8% of the participants chose (I will wait and see its effects on others) – many of whom could have possibly chosen (Yes) or (Agree) had the waiting choice been eliminated. A systematic review noted that 'When answer options included different timings for vaccination, more people chose to wait than get it as soon as possible' and that the two answer choices (Yes/No) received relatively high affirmatives (*Lin et al., 2020*). Only 3 of about 70 studies and polls in this

review included the choice of 'wait a while until others have taken it' for the question of vaccine acceptance. In these three polls, conducted before vaccine availability, the percentage of those who chose that they would take the vaccine as soon as they can (or as soon as possible) was low (21–28%). This indicates that vaccine acceptance may be overestimated in many studies and highlights the need to redefine vaccine acceptance in a uniform way among different studies.

Another factor that may explain the lower rate of acceptance observed in the study is the nature of our survey population. Social and political differences were found to have a prominent effect on COVID-19 vaccine acceptance, especially that many people assumed political interference in the vaccine and in the pandemic itself (*Lin et al., 2020*). Vaccine acceptance was lower in Arab countries in previous studies: Jordan (37.4%) (*El-Elimat et al., 2020*), Saudi (64.72%) (*Al-Mohaithef and Padhi, 2020*) and (44.7%) (*Magadmi and Kamel, 2020*), and in a small multinational survey that included several countries, mainly Jordan and Kuwait (29.4%) (*Sallam et al., 2021*).

The chronological analysis of vaccine acceptance and time – in local and multinational studies – does not show a linear relationship – if anything, the public's acceptance can be best described as fluctuating. Several surveys conducted in the last third of 2020 have shown a decrease in vaccination acceptance compared with previous surveys (*Lin et al., 2020*; *Ipsos, 2020*; *Wang et al., 2021*; *Kreps et al., 2020*). For example, the intent to vaccinate has declined in 10 of the 15 countries from August to October 2020 (*Ipsos, 2020*). A systematic review of publications until 20 October 2020 showed declining vaccine acceptance (from >70% in March to <50% in October) with demographic, socioeconomic, and partisan divides observed (*Lin et al., 2020*). However, a more recent multinational survey conducted from November 2020 to mid-January 2021 in 15 countries (*Imperial College London, 2021*) and a study conducted from 28 to 31 January 2020 in 14 countries *Ipsos, 2021* have both shown an increase in vaccine acceptance.

As for Arab countries, the more recent studies (*Sallam et al., 2021*; *El-Elimat et al., 2020*; *Hussein, 2021*) show lower acceptance rates than the earlier ones. For example, A study conducted in Egypt (n = 559) during March 2020 found out that about 73.0% were looking forward to getting the vaccine when available (*Al-Mohaithef and Padhi, 2020*; *Magadmi and Kamel, 2020*; *Abdelhafiz et al., 2020*). However, the more recent study on HCW in Egypt (n = 496) during December 2020 concluded that only 13.5% totally agree to receiving the vaccine and 32.4% somewhat agree (*Hussein, 2021*). In our survey that is more recent than the mentioned studies, 17.0% and 24.0% of participants in Egypt (general [n = 5339] and HCW [n = 1250], respectively) were willing to take the vaccine.

A Saudi Arabia study published in May 2020 observed that 64.7% of participants (n = 1000) showed interest to accept the COVID-19 vaccine if it is available (*Al-Mohaithef and Padhi, 2020*), and another study conducted during May (n = 3101) showed a 44.7% acceptance rate (*Magadmi and Kamel, 2020*). However, in a more recent study in Saudi Arabia too, published in December 2020, 31.8% of participants (n = 154) showed acceptance (*Sallam et al., 2021*). In our survey that is more recent than the mentioned studies, 19.8% of participants from Saudi Arabia (n = 3588) showed acceptance.

Health care workers (HCWs) are at increased risk of acquiring and transmitting COVID-19 infection. Moreover, they present role models for communities with regards to attitudes towards COVID-19 vaccination. Therefore, vaccine hesitancy in this group is of a special concern, and we discussed hesitancy in Arab healthcare workers elsewhere (*Qunaibi et al., 2021*).

This study comes with few limitations. Similar to several previous surveys (*Lin et al., 2020*), participants were recruited through social media. Being an online survey, our study may have under-represented certain groups of individuals, including members of older age groups and those who are not active on social media. We cannot rule out selection bias that might have affected our results. Other high-risk groups such as people with chronic diseases are well represented (n = 5839) or even over-represented (HCW, n = 5708). Our sample size was not pre-planned but was rather arbitrary reflecting a convenience sample. We believe that the large number of participants and the consistency of results in different countries that were geographically close and similar socioeconomically confirm the reliability of our survey.

## Conclusion

Our results show high COVID-19 vaccine hesitancy among Arab respondents residing inside and outside the Arab world after millions of people around the world have received the vaccine. The main

reasons for hesitancy are concerns about safety and distrust in health care policies, vaccine expedited production, and published studies, with the distrust being notably higher among respondents residing in the Arab countries. Given that the vaccine is being purchased from state expenditure, the high vaccine hesitancy could further compromise the economies of Arab countries in addition to the pandemic health hazard. At the same time, mandating the vaccine is not a desirable choice and could further increase the distrust. With the highly dynamic nature of the pandemic and vaccine production process and the interplay of ever-changing factors that affect vaccine acceptance, our study needs to be replicated at a later time to measure the change in public acceptance. The high proportion of people willing to wait until others have received the vaccine and the unavailability of the preferred vaccine for others show a need to create a uniform definition for vaccine acceptance in the surveys to avoid misestimation.

## Additional information

### Funding
No external funding was received for this work.

### Author contributions
Eyad A Qunaibi, Conceptualization, Data curation, Supervision, Investigation, Methodology, Writing - original draft, Literature survey; Mohamed Helmy, Software, Writing - original draft, Writing - review and editing; Iman Basheti, Conceptualization, Formal analysis, Validation, Methodology, Writing - review and editing; Iyad Sultan, Conceptualization, Data curation, Formal analysis, Investigation, Visualization, Methodology, Writing - original draft

### Author ORCIDs
Eyad A Qunaibi (iD) https://orcid.org/0000-0003-0648-0757
Mohamed Helmy (iD) https://orcid.org/0000-0002-9561-7956
Iyad Sultan (iD) https://orcid.org/0000-0002-2664-1565

### Ethics
Human subjects: This study was approved by the institutional review board (IRB) at King Hussein Cancer center (Approval No. 21 KHCC 0.34).

### Decision letter and Author response
Decision letter https://doi.org/10.7554/eLife.68038.sa1
Author response https://doi.org/10.7554/eLife.68038.sa2

## Additional files

### Supplementary files
• Supplementary file 1. Survey of the attitudes of Arabs towards COVID-19 vaccines .

• Supplementary file 2. Differences in barriers for acceptance according to country of residence, gender and academic achievement, compared using chi-square test.

• Reporting standard 1. Checklist of items that should be included in reports of cross-sectional studies.

• Transparent reporting form

### Data availability
All data are available at our study website: https://mainapp.shinyapps.io/CVHAA. All the project's data and code has been deposited on Github. *E. Qunaibi* (2021) COVID-Vaccine-Arab-Survey. GitHub. https://github.com/MoHelmy/COVID-Vaccine-Arab-Survey. 5d1a881 (copy archived at https://archive.softwareheritage.org/swh:1:rev:5d1a8814d9dc1e3a87fb873ef994f9e0af2fce98).

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

## Appendix 1

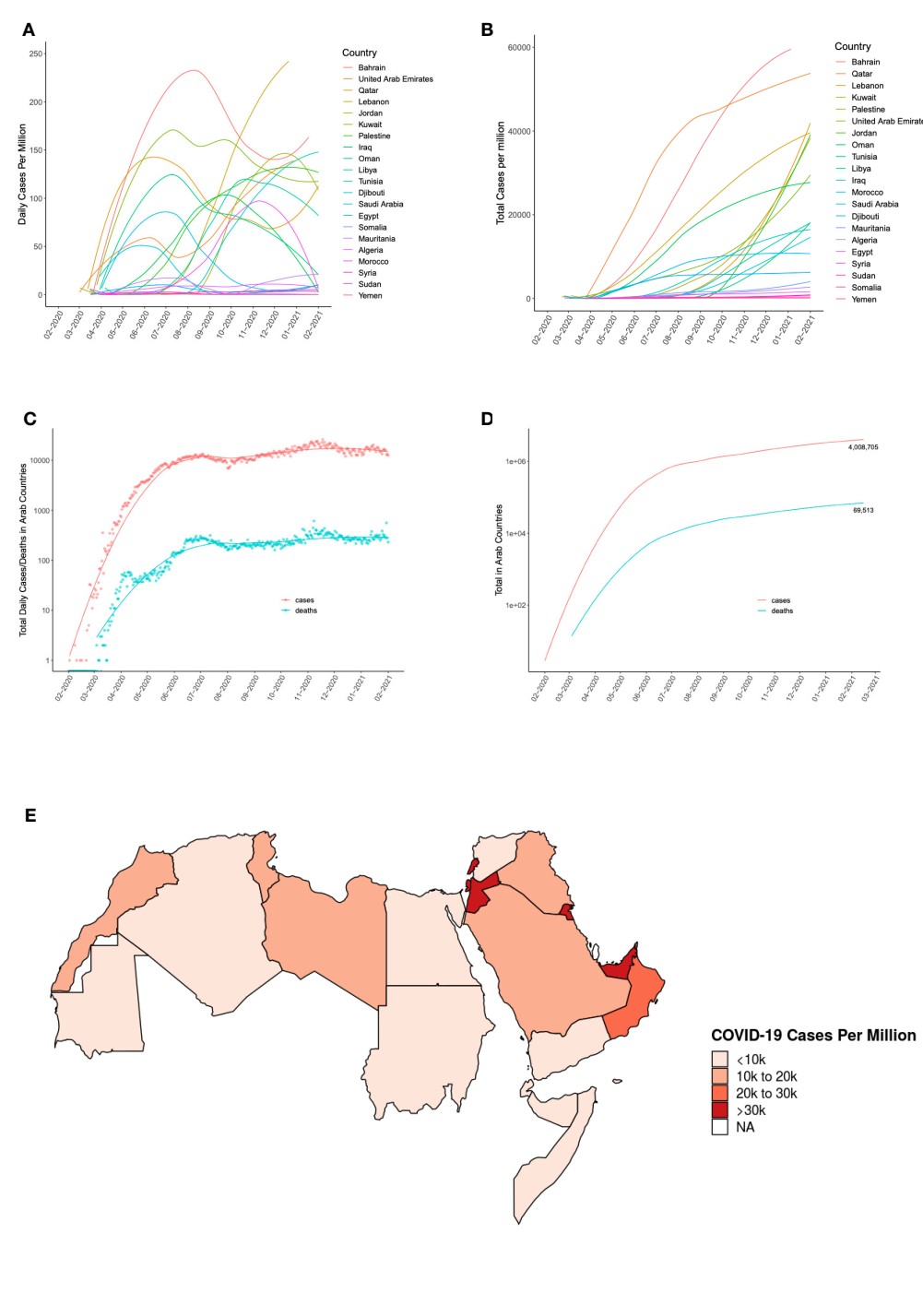

**Appendix 1—figure 1.** The burden of COVID-19 outbreak on Arab countries with panels representing (A) the daily cases per million in different countries, (B) cumulative cases per million in

each country, (C) total daily cases and deaths in all Arab countries, (D) cumulative number of confirmed cases and deaths in all Arab countries, and (E) a map showing the differences in total cumulative confirmed cases per million capita.

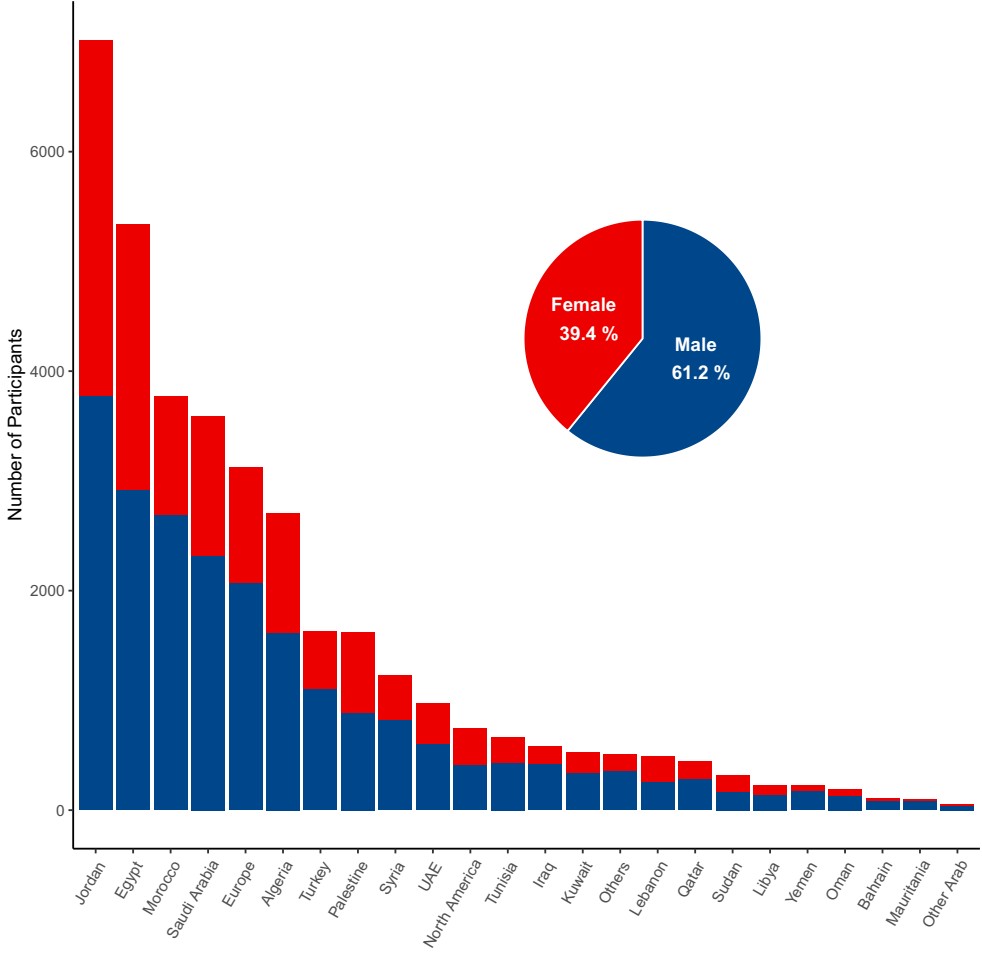

**Appendix 1—figure 2.** A bar plot and a pie chart showing distribution of participants according to country of residence and gender.

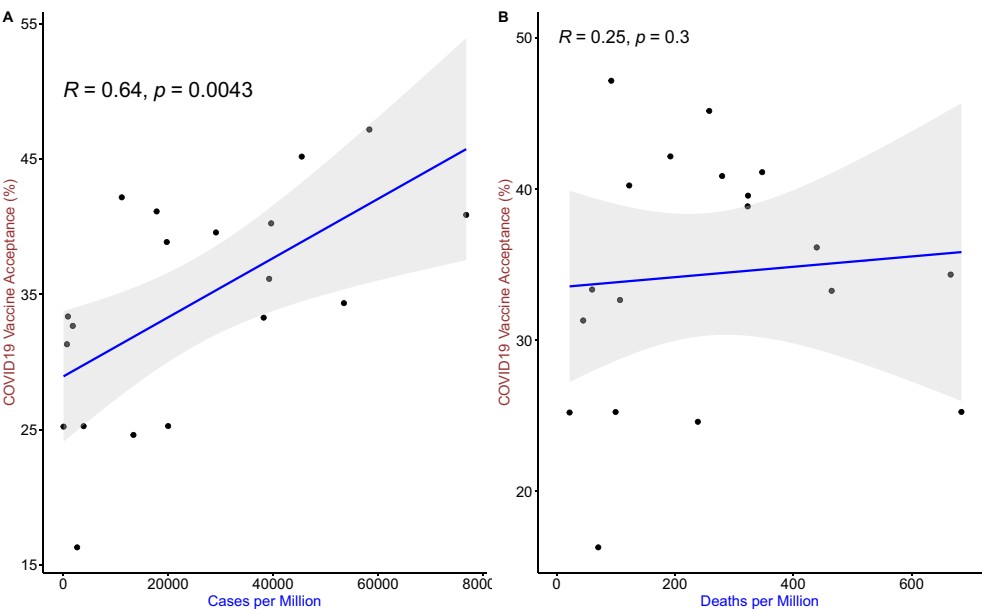

**Appendix 1—figure 3.** Scatter plots showing correlation between vaccine acceptance in 36,220 survey participants and (**A**) the total number of confirmed COVID-19 cases per million and (**B**) the total COVID-19-related deaths per million in Arab countries.

