## [Decision Letter]

**Acceptance summary:**

The current SARS Cov-2 pandemic can only be effectively controlled by the delivery of effective vaccines to non-immune populations but a significant hurdle to this is vaccine hesitancy and the consequent failure to take up the offer of vaccination. To counter this public health authorities need to have robust data on the extent of vaccine rejection and the underlying reasons for this. The paper by Qunaibi and colleagues addresses this question in a large scale online study of over 36,000 participants from all the Arab speaking countries plus > 100 other countries with Arabs living in them. The data represent the largest study to date and the detailed questionnaires provide context behind the acceptance or rejection of a highly effective vaccine against a life threatening infectious disease. As one of the reviewers comments the study "provides important dataset for the health authorities in Arab and non-Arab countries to plan mitigation strategies to improve vaccine acceptance".

**Decision letter after peer review:**

Thank you for your submission entitled "A High Rate of COVID-19 Vaccine Hesitancy in a Large-scale Survey on Arabs". Your paper has been evaluated by two reviewers and by a Reviewing Editor and a Senior Editor. As is customary in *eLife*, the reviewers have discussed their critiques with one another and with the editors. We are pleased to inform you that your article has been accepted for publication in *eLife*.

Please take note of the points below, in case you may wish to revise the paper prior to production. We hope you will continue to support *eLife*.

*Reviewer #1:*

The authors conducted a large-scale online survey that covers the region of the Arab countries and the Arabic-speaking people residing outside the Arab region. The main focus of the survey is to assess the vaccination acceptance in the Arab region accurately and in details. The work has multiple strength points:

1. The large sample size. The authors successfully recruited over 36K participants which makes this survey the largest worldwide, to date.

2. The detailed differentiation between the levels of acceptance/hesitation came in 5 different detailed levels, unlike other surveys that make them from 1 to 10 or yes/no.

3. The very detailed reasoning of the vaccine hesitancy. The authors asked the participants to detail their reasons for hesitation or rejection by giving them 29 different reasons that cover almost all possible hesitancy reasons.

4. The study is the first multinational study in the region. The Arab region was not studied at all, except a few local studies with limited participation. So, this study comes to fill a gap in the data about the region.

The results of the survey show a high rate of vaccination hesitancy, which comes in agreement with the escalated rates worldwide post-vaccination campaigns and news about unexpected side effects.

*Reviewer #2:*

The manuscript "A High Rate of COVID-19 Vaccine Hesitancy 1 in a Large-scale Survey on Arabs" submitted by Eyad A. Qunaibi presents detailed description on hesitancy against COVID-19 vaccine with thorough analyses of factors influencing such hesitancy across the entire Arab world. The analysis conveniently included a small, yet reasonably-sized, sample of Arabs living in non-Arab countries, which provided a contrasting substance to compare with the main study. The manuscript is helpful in understanding the spectrum of barriers limiting people's acceptance of vaccination against COVID-19, and further highlights major barriers consistent with previously published literature, yet at a much larger scale.

Main barriers cluster around lack of knowledge that likely results in mistrust and worry. This is highlighted in the high proportion of participants not receiving the annual influenza vaccine and the concerns with the safety of COVID-19 particularly due to the speedy emergency use authorization of such vaccines.

One major value is the thoroughness of barriers included in the manuscript, which provides important dataset for the health authorities in Arab and non-Arab countries to plan mitigation strategies to improve vaccine acceptance.

The only optional recommendation that may improve on the quality of the manuscript, if data is available, is to understand similarities and differences in response to barriers between countries that show higher vs. lower hesitancy to COVID-19 vaccine. For example, was there a response pattern among Qatar, Oman, and Kuwait that clusters uniquely compared with response pattern among Tunisia, Algeria, and Morocco? Similarly, were there similar pattern of responses to barriers in countries with high acceptance such as in Qatar, Oman, and Kuwait compared with non-Arab countries of match acceptance such as North America? Such data may provide additional guiding details that will allow health authorities to devise more accurate policies to minimize such country-specific barriers and improve the rate of vaccine acceptance.

---

## [Author Response]

Reviewer #2:[…] The only optional recommendation that may improve on the quality of the manuscript, if data is available, is to understand similarities and differences in response to barriers between countries that show higher vs. lower hesitancy to COVID-19 vaccine. For example, was there a response pattern among Qatar, Oman, and Kuwait that clusters uniquely compared with response pattern among Tunisia, Algeria, and Morocco? Similarly, were there similar pattern of responses to barriers in countries with high acceptance such as in Qatar, Oman, and Kuwait compared with non-Arab countries of match acceptance such as North America? Such data may provide additional guiding details that will allow health authorities to devise more accurate policies to minimize such country-specific barriers and improve the rate of vaccine acceptance.

Thank you for the nice recommendation. Per the reviewer’s comment, we compared the response to barriers between countries that show higher hesitancy (Algeria, Mauritania, Tunisia) vs. lower hesitancy (Kuwait, Oman, Qatar) to COVID-19 vaccine. We found that the most prominent difference was in choosing the following barrier:

“I do not trust the healthcare policies applied in my country”.

The response rate to this barrier was as follows:

Countries with high hesitancy: Algeria 51.7%, Mauritania 46.5%, Tunisia 38.8%.

Countries with low hesitancy: Kuwait 18.5%, Oman 16.6%, Qatar 4.7%.